# Polymer-dispersed liquid crystal elastomers as moldable shape-programmable material

Matej Bobnar[1], Nikita Derets[1,2], Saide Umerova[1], Valentina Domenici[3], Nikola Novak[1], Marta Lavrič[1], George Cordoyiannis[1], Boštjan Zalar[1,4] & Andraž Rešetič ●[1] ✉

The current development of soft shape-memory materials often results in materials that are typically limited to the synthesis of thin-walled specimens and usually rely on complex, low-yield manufacturing techniques to fabricate macro-sized, solid three-dimensional objects. However, such geometrical limitations and slow production rates can significantly hinder their practical implementation. In this work, we demonstrate a shape-memory composite material that can be effortlessly molded into arbitrary shapes or sizes. The composite material is made from main-chain liquid crystal elastomer (MC-LCE) microparticles dispersed in a silicone polymer matrix. Shape-programmability is achieved via low-temperature induced glassiness and hardening of MC-LCE inclusions, which effectively freezes-in any mechanically instilled deformations. Once thermally reset, the composite returns to its initial shape and can be shape-programmed again. Magnetically aligning MC-LCE microparticles prior to curing allows the shape-programmed artefacts to be additionally thermomechanically functionalized. Therefore, our material enables efficient morphing among the virgin, thermally-programmed, and thermomechanically-controlled shapes.

A great number of emerging material-related technologies strive to enhance commonplace materials with additional functionalities in order to expand their usage beyond the fundamental scope. By combining the inherent characteristics of several constituents into one multifunctional material, many devices that rely on intricate assemblies of passive components to complete the intended task can be simplified or even completely substituted with a single active element. Moreover, introducing exotic new properties in otherwise conventional materials should make it possible to realize previously unfeasible tasks. Soft shape-morphing materials are typical such representatives, which have the ability to deform or change shape when subjected to an external stimulus. The development of such materials is essential for the advancement of soft robotics[1–4], as well as towards applications from soft actuators[5–9], to sensors[10,11] and many more[12–16].

Currently, the most promising representatives of soft morphing materials are shape-memory (SM) polymers[17] and liquid crystal elastomers (LCEs)[18,19]. SM polymers are shape-reprogrammable soft materials that mostly rely on the formation and dissolution of local crystalline clusters in their polymer network to freeze-in and relax imposed deformations[17]. LCEs, on the other hand, combine the elastic properties of the cross-linked network with the orientational ordering of liquid crystals (LCs), i.e. the mesogen molecular components. The temperature-initiated collapse and regeneration of the LC order drive the polymer network of LCEs between the strained and initial configuration, effectively deforming the specimen between the two states[20].

[1]Jožef Stefan Institute, Solid State Physics Department, Jamova cesta 39, 1000 Ljubljana, Slovenia. [2]On leave from: Ioffe Institute, Division of Physics of Dielectrics and Semiconductors, Politekhnicheskaya 26, 194021 St. Petersburg, Russia. [3]Dipartimento di Chimica e Chimica Industriale, Università degli studi di Pisa, via Moruzzi 13, 56124 Pisa, Italy. [4]Jožef Stefan International Postgraduate School, Jamova cesta 39, 1000 Ljubljana, Slovenia. ✉e-mail: andraz.resetic@ijs.si

While considered as potentially useful, the shape-memory and actuating capabilities of morphing materials are yet to be practically implemented. We argue that some of the major obstacles preventing the application of these materials are the geometrical limitations posed by their synthesis procedures. These typically require vast amounts of solvent that needs to be removed once the reaction is completed[21,22]. Molding is therefore performed on swelled systems, which upon drying do not retain the initial geometry of the mold. Photo-curing is also commonly used for the production of SM material[7,8,23,24], but the final geometries are limited by the short penetration lengths of UV-light. Together with other similar methods[25–27], any larger-scale fabrication and molding of SM materials remains generally restricted to thin-film geometries. On the other hand, additive manufacturing[28–31] and similar techniques[16,32] are used to fabricate 3D solid specimens with intriguing shapes and actuations, but these require sophisticated equipment and lack in scalability for any larger production output. Moreover, the actuation of such 3D-printed specimens so far mostly resembles bending into thin-walled geometries rather than actual morphing between 3D solid objects, which can again limit their practical utilization[29,30]. A material production method that can bridge the gap between the two limits, i.e. the least demanding, but geometrically restricted synthesis versus the low-yield and complex 3D printing, should help in accelerating the implementation of soft morphing materials.

In this paper, we introduce a practical approach of obtaining a moldable shape-memory material that can be effortlessly cast into specimens of arbitrary shape and size, and exhibits temperature-controlled morphing between solid 3D objects. We previously demonstrated that the doping and magnetically-induced ordering of LCE microparticles in a poly(dimethylsiloxane) (PDMS) matrix renders the composite material thermomechanically (TM) active[33]. Such soft-soft composites, referred to as 'polymer-dispersed liquid crystal elastomers' (PDLCEs), exhibit a reversible contraction in the direction of the permanently magnetically imprinted TM anisotropy. In the present study, we utilize the intrinsic glassiness of nematic MC-LCE microparticles, rather than their TM actuation capabilities, to functionalize the composite material with reprogrammable shape-memory that resemble those of SM polymers and related composites[9,34,35]. The temperature-dependent Young's modulus of the MC-LCE inclusions is exploited for the efficient control of elastic relaxations of the surrounding matrix. PDLCEs can therefore be reprogrammed into new geometries through thermal and mechanical manipulation, as their memorized shape fully resets once heated above the glass transition temperature. The pre-polymerized PDLCE resin thermally cures without volume change, which makes it suitable for molding, specifically into bulk, solid 3D specimens of defined shapes. No external magnetic field is needed for SM functionalization of the composite, however, the magnetic alignment of LCE particles during the PDLCE fabrication process allows an additional reversible shape-change to be introduced, via the temperature-driven nematic to isotropic phase transition[33]. Combined with the SM programmability, the material can be formed into various morphing configurations exhibiting a two-step, temperature-driven shape transformation.

## Results

### Shape memory in main-chain LCEs

The main bulk properties of a composite system stem from the combination of individual physical traits of its components. By dispersing dopants into a host material, one can induce new functionalities and attributes[36,37]. This notion has been adopted in our previous study, by functionalizing conventional silicone elastomers with TM properties via magnetically oriented, TM active LCE micro-inclusions[33]. In our current work, the same procedure is used for the purpose of instilling shape-programmability into the elastomer, which is achieved by

utilizing the intrinsic SM properties found in the MC-LCE dopant material (Fig. 1).

The MC-LCE crosslinked polymer network (Fig. 1a) consists primarily of main-chain nematogens (MC-LCs) with a high-temperature-persistent glass phase ($T_{Glass-Nem} = 379$ K) and a high-temperature nematic (Nem) to isotropic (Iso) phase transition ($T_{Nem-Iso} = 437$ K, Supplementary Fig. 1 and Supplementary Fig. 2). MC-LCEs can be synthesized as either strip-shaped monodomain samples using a two-step synthesis procedure[38] or as bulk-sized, partially ordered specimens requiring a single-step synthesis in a strong magnetic field[33,39] (Fig. 1b, please refer to Experimental Procedures section for additional details on sample preparation). The monodomain strip-shaped MC-LCE exhibits up to $\lambda_{MC-LCE} = 91\%$ contraction if heated above $T_{Nem-Iso} \approx 396$ K (Supplementary Fig. 3 and Supplementary Fig. 4). The thermomechanical coefficient of the material is determined as $\lambda = (L_0 / L_{ref} - 1) \times 100$, where $L_0$ is the initial length measured at $T = 300$ K and $L_{ref}$ is the actuated length observed at $T = 430$ K. Each phase transition is also accompanied by a significant change of the material's mechanical properties[40–42]. In our particular case, the Young's moduli between the 'harder' glassy and the 'softer' nematic phase differ for more than two orders of magnitude (Fig. 1c). Individual stress/strain measurements behind the temperature dependent Young's modulus values, $E_{MC-LCE}(T)$, (Supplementary Fig. 5) show a wide hysteretic response in the Glass-Nem transition temperature region (between 310 K and 360 K). Such hysteretic strain relaxation is typical for LCEs exhibiting soft-elastic domain reorientations after the stress is removed[43–45]. No hysteresis is observed below 310 K, since the high glassiness prevents the domain alignment. This increase in domain rigidity at lower temperatures combined with the relatively high glass-to-nematic phase transition temperature ($T_{Glass-Nem} = 325$ K, Supplementary Fig. 4) is crucial for the establishment of a room temperature stabilized SM[35,46,47]. With an appropriate temperature and mechanical manipulation, i.e. by cooling a strained MC-LCE from the isotropic to the glass phase (thermal-cycling), the shape of the specimen can be effectively programmed.

We demonstrate the aforementioned by means of: (a) tensile stress programming of a strip-shaped monodomain sample (Fig. 1d left) and (b) compressive stress programming of a partially magnetically ordered MC-LCE cube (Fig. 1d right). In both cases, the programming thermal cycles (full red and blue circles) show a significant change in length above $T_{Glass-Nem} = 325$ K, as the system enters the much softer nematic phase. Upon further heating, the samples continue to TM contract and settle at the minimum value after the isotropic phase is established. Upon cooling, the strain values begin to diverge from the heating curve at 360 K and relax to a new stable value at room temperature upon load removal. The fixation ratio $R_{fix} = (\varepsilon_{mem} - \varepsilon_{prog})/\varepsilon_{prog} + 1$ is introduced to quantify the amount of strain remaining after the programming thermal cycle is completed and the sample unloaded. Here, $\varepsilon_{mem}$ is the amount of memorized and $\varepsilon_{prog}$ the amount of programmable strain, i.e. the strain values corresponding to points 3 and 4 in Fig. 1d, respectively. The corresponding $R_{fix}$ (Supplementary Table 1) indicates that no relaxation occurs during tensile programming ($R_{fix} = 0.99$), while up to 80% of strain is memorized when programming with compressive stress ($R_{fix} = 0.79$). For both cases, the programmed states were preserved for more than 6 months. During the relaxation thermal cycle, i.e. thermal-cycling without an applied load ($\sigma_{Load} \rightarrow 0$), the programmed strain begins to relax in the nematic phase and the sample completely resets to its initial length once cooled to room temperature.

These measurements show that the majority of programmable strain is generated at the outset of the nematic phase ($T \approx 360$ K), at which the domain alignment becomes most susceptible to mechanical stress. This is even more evident in the polydomain MC-LCEs where a large spontaneous elongation is observed around the same temperature (Supplementary Fig. 6). However, due to the particularly high

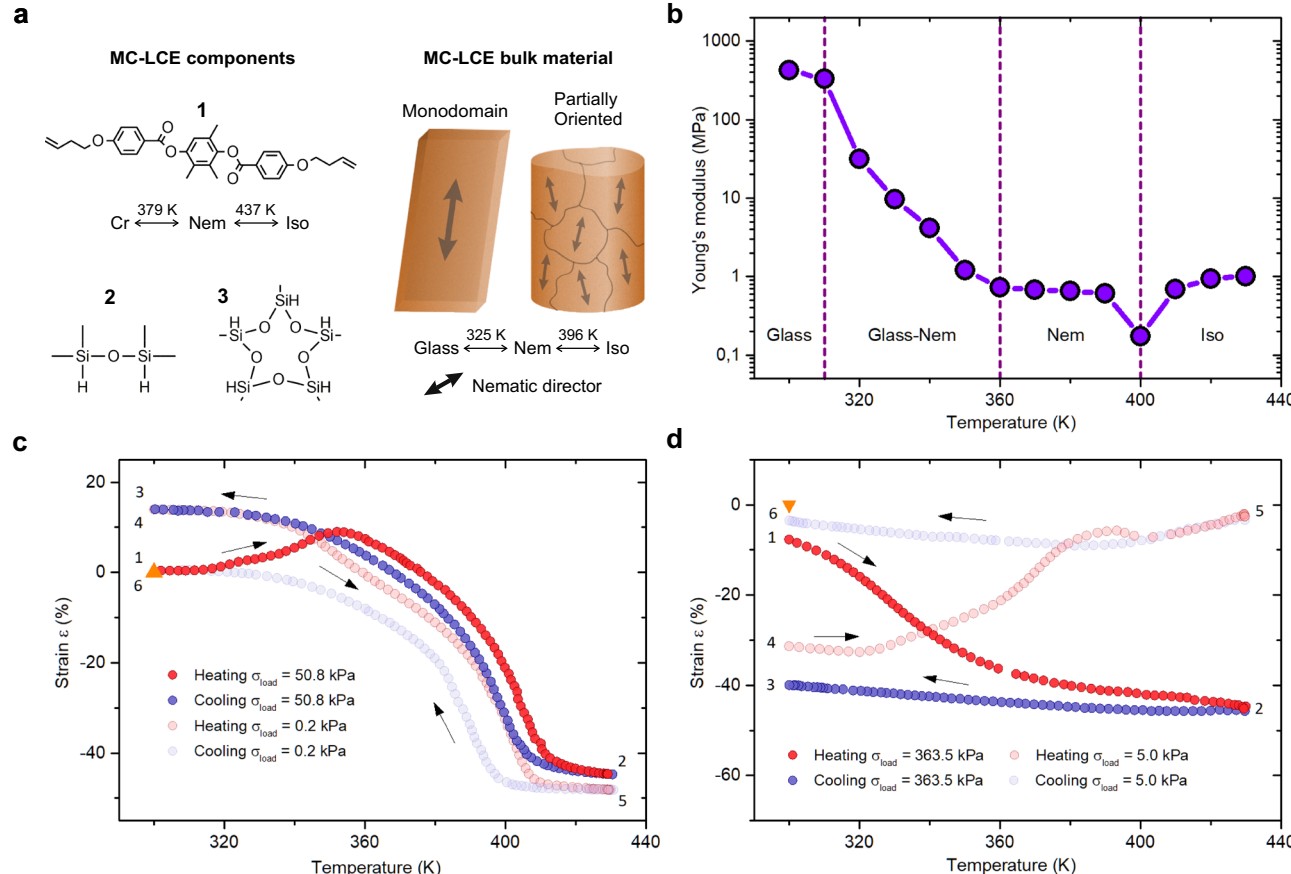

**Fig. 1 | MC-LCE material composition and thermomechanical properties.**
**a** Chemical composition of the MC-LCE material components: main-chain mesogenic monomer MC-LC−1, polymer chain extender−2 and five-points crosslinker−3. MC-LCE material was synthesized as a monodomain strip or as a partially magnetically ordered specimen. Strip shaped MC-LCEs were only used for measurements, while partially ordered MC-LCEs also served as parent material for the production of PDLCEs. See Experimental procedures section for a detailed synthesis procedure. **b** Temperature-dependent Young's modulus ($E_{MC-LCE}(T)$) of a monodomain MC-LCE specimen changes significantly with temperature and is indicative of the liquid crystalline phase currently present in the system[40–42,54]. Based on the $E_{MC-LCE}(T)$ profile, we can identify all three liquid crystal phases (glass, nematic and isotropic), along with transition temperatures indicated by the

anomalies in the $E_{MC-LCE}(T)$ values. Tensile (**c**) and compression (**d**) stress programming of the MC-LCE material is demonstrated by thermal-cycling of the material under a constant applied load (points 1–3, full circles). Upon load removal, the imposed strain relaxes to a new stable value (4). The memorized strain is then reset with another thermal cycle under a nominal load, $\sigma_{Load} \rightarrow 0$, upon which the sample returns to its initial length (4–6, empty circles). Tensile stress programming was performed on a strip-shaped monodomain MC-LCE, whereas a small rectangular specimen cut from the partially ordered MC-LCE was used for compressive programming. In both cases, the stress was applied in the direction of the imprinted nematic ordering. Orange triangles denote the sample's length/thickness before thermal-cycling.

sensitivity of polydomain systems to soft-elastic ordering, it could be difficult to fully reset the memorized strains, since even a minimal applied load is enough to render them monodomain (see empty blue line in Supplementary Fig. 6).

## Shape-programmable main-chain PDLCEs

The MC-LCE material can be synthesized as sizeable 3D specimens with good SM functionality. However, the high solvent content required for the synthesis demands a complex and precise control of solvent evaporation and polymerization parameters to produce any kind of macro-sized systems with well-defined geometries. On the contrary, by employing MC-LCEs as microparticles, it is feasible to prepare a PDLCE mixture that is easily molded into arbitrary shapes and produced in bulk quantities. No particle orientational order is required for SM functionalization, in contrast to functionalization with TM properties[33,39]. The particles function as malleable and temperature sensitive 'solidifiers' that prevent strain relaxation of the surrounding elastic matrix. The use of external orientational fields or any other intricate techniques for instilling molecular anisotropy is therefore not necessary, making it possible to produce SM capable PDLCEs using only rudimentary polymer manufacturing methods.

PDLCEs used in this study are composed of $w_{MC-LCE} = 0.40$ MC-LCE microparticle mass concentration dispersed in Sylgard® 184 Silicone elastomer polymer (see Experimental procedures section). The liquid PDLCE mixture is produced by freeze-fracturing partially magnetically-ordered MC-LCE material combined with liquid PDMS (Fig. 2). To avoid any incomplete relaxations of programmed deformations in the PDLCEs, which could be caused by highly stress-sensitive polydomain inclusions, only high-yield magnetic synthesis was used for the microparticles' production. Such MC-LCE microparticles are considered to be essentially monodomain, as they are shown to retain the same mesophase behavior and TM properties as their monodomain bulk equivalents[33,39,40]. The prepared dispersion is then cast in a polytetrafluoroethylene (PTFE) mold of any desired shape, evacuated and thermally cured into PDLCE composites. Once polymerized, the PDLCEs can be reshaped in the same way as MC-LCEs (Fig. 2b).

Shape-programming and relaxation of a cylindrical PDLCE upon thermal cycling is presented in Fig. 3a. It can be observed that the overall SM characteristic of the parent MC-LCE material is well adopted by the PDLCE composites. Differences are seen in the minimal TM response (<2%) of the PDLCE due to the lack of particle orientational

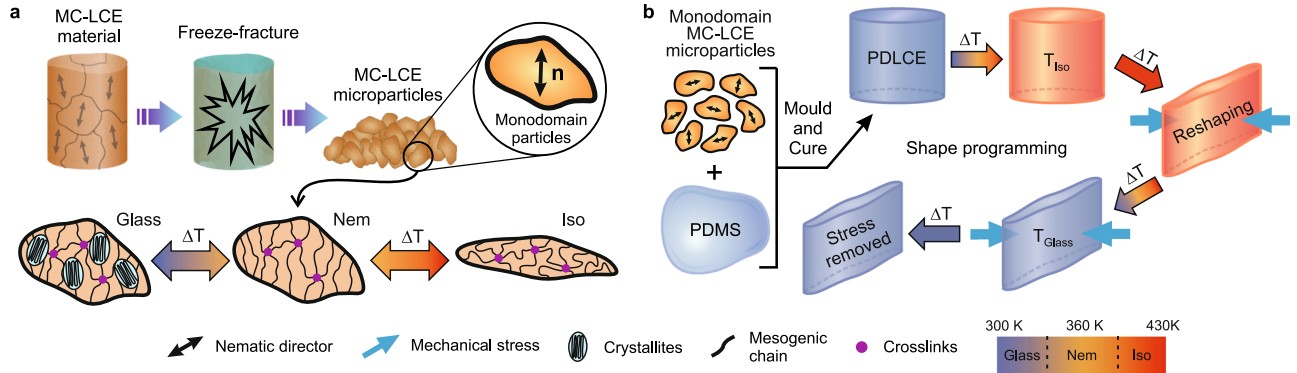

**Fig. 2 | Schematics of PDLCE synthesis and shape-programming. a** Freeze-fracturing yields MC-LCE microparticles from the magnetically ordered and polymerized MC-LCE material. The obtained MC-LCE microparticles are significantly smaller in size than magnetically formed nematic domains, thus, they procure a monodomain mesogenic structure. The same mesophases are present in the MC-LCE microparticles as in the parent material, i.e. a glassy phase, followed by the nematic and isotropic phases as temperature increases, accompanied by a significant contraction along the particle's nematic director. **b** PDLCEs are prepared from a suspension of MC-LCE microparticles in PDMS, which is then cured and molded into a desired shape. A new shape can then be programmed by reshaping the composite when above $T_{Iso}$ and cooling it to $T_{Room}$ while maintaining the shape. Once cooled, the programmed shape is memorized.

order and in the case of reduced amount of memorized strain ($R_{fix} \approx 0.81$, Supplementary Table 1). This is to be expected, since the much softer and elastic PDMS matrix, which comprises most of the composite ($w_{PDMS} = 60\%$), does not exhibit any SM-like properties (Supplementary Fig. 7).

The PDLCEs also mirror the thermomechanical properties of MC-LCEs, with the overall Young's modulus values reduced towards those of the PDMS matrix. These values can be customized by choosing the hardness of the matrix (Fig. 3b). Increasing the Young's modulus of the PDMS, raises also the fixation ratio of the programmed deformations (Fig. 3b), as a stiffer matrix results to much lower imposed strains in the composites (Fig. 3c).

Once programmed, the memorized strains are preserved and can be reset by heating the sample. Interestingly, the deformations relax well before $T_{Nem-Iso}$. If a higher programmable stress is applied, the sample resets above the isotropic temperature (Supplementary Fig. 8), which suggests that the relaxation dynamics could be influenced by the instilled degree of liquid crystal order in the LCE microparticles. Further detailed investigations are needed to provide solid evidence on this.

Due to the typical 2D nature of SM materials, compressive stress programming of SM material is typically restricted to surface-embossing of thin samples, which results in micro-scaled deformations of the material's thickness[8,25,48,49]. Since PDLCEs can be molded into voluminous, solid 3D objects, a compressive programmable stress can be used to imprint substantial deformations into the specimen (up to $\varepsilon_{SM} = 49\%$, Supplementary Fig. 8). When compared to tensile stress programming, the fixation ratio of compressive stress programming is similar ($R_{fix} \approx 0.80$, Supplementary Table 1), but the memorized strain resets only in the isotropic phase. However, it will be demonstrated that compressive strains do relax in the nematic phase when the samples are not mechanically loaded during thermal reset. Since a nominal, constantly applied stress is necessary to properly probe the specimen's length, it is reasonable to assume that even such small loads can sustain a high degree of imposed LC order, as observed in polydomain MC-LCEs (Supplementary Fig. 6).

In addition to the basic tensile and compressive stress programming, other deformation modes can be applied, such as bending or torsional shear; the latter exhibiting rotational relaxation (Fig. 4a). Moreover, the surface of a PDLCE specimen can be easily embossed or debossed (Fig. 4b), making this material particularly useful for surface templating or as an impressioning tool. Bulk, three-dimensionally produced PDLCE samples can perform morphing between solid objects. For instance, a solid spherical PDLCE can be programmed into

a cube, which in turn morphs back into the original spherical shape once thermally reset (Fig. 4c). The same PDLCE can then be reprogrammed into a new stable shape, in this case a tetrahedron, and reset when desired. Here, shape-morphing occurs via relaxations of the material's deformed solid volume. Due to the general mechanical robustness of such solid systems, such an SM material can be superior to its 2D counterparts, especially when it comes to applications. Potential implementations of the material can range from active rubber sealants and mechanical energy storage devices to soft actuators and mechanical switches, to name just a few. Moreover, substantially thick samples can be prepared with molding or extrusion processes, instead of resorting to impractical and time-consuming layering techniques.

## Shape-memory coupled with thermomechanical actuation

As demonstrated, the memorized deformations in a PDLCE can relax at significantly lower temperatures than the nematic-isotropic phase transition temperature of the incorporated MC-LCE particles. This makes it possible to further functionalize the PDLCEs with an additional TM response which is thermally well-separated from the SM relaxations. Such TM active PDLCEs exhibit a two-step shape change when heated from room temperature to the isotropic phase. To achieve any significant TM actuation, the MC-LCE microparticles have to be oriented so that the particles' associated nematic directors are uniformly aligned (Fig. 5a). This ensures that particles contract in the same direction and collectively initiate a substantial deformation of the composite. In order to fulfill this condition an external magnetic field has to be applied on the pre-polymerized PDLCE dispersion until the particle orientational order is formed along the field direction. By curing the surrounding matrix, the generated TM anisotropy is permanently imprinted into the composite[33,39]. With the optimum composition of $w_{MC-LCE} = 0.40$, the TM output of a magnetically ordered PDLCE has a maximum value of $\lambda_{max} = 20.3\%$ (Fig. 5b). The PDLCE's TM properties can be further customized by changing the particle concentration (Fig. 5c) or tuning the matrix' hardness (inset in Fig. 5b).

Apart from the noticeable TM contraction, the introduced anisotropy in magnetically ordered PDLCEs has no influence on the shape-programming process, as it can be seen in Fig. 5d, left graph (steps 1 to 3). The fixation ratios remain comparable as well ($R_{fix} \approx 0.80$, see Supplementary Table 1), showing that the imprinted particle alignment has little to no effect on the mechanical shape-programming. Moreover, two well-separated thermomechanical responses are observed during thermal relaxation in the case of tensile stress programming (steps 4 to 5). Here, the specimen first

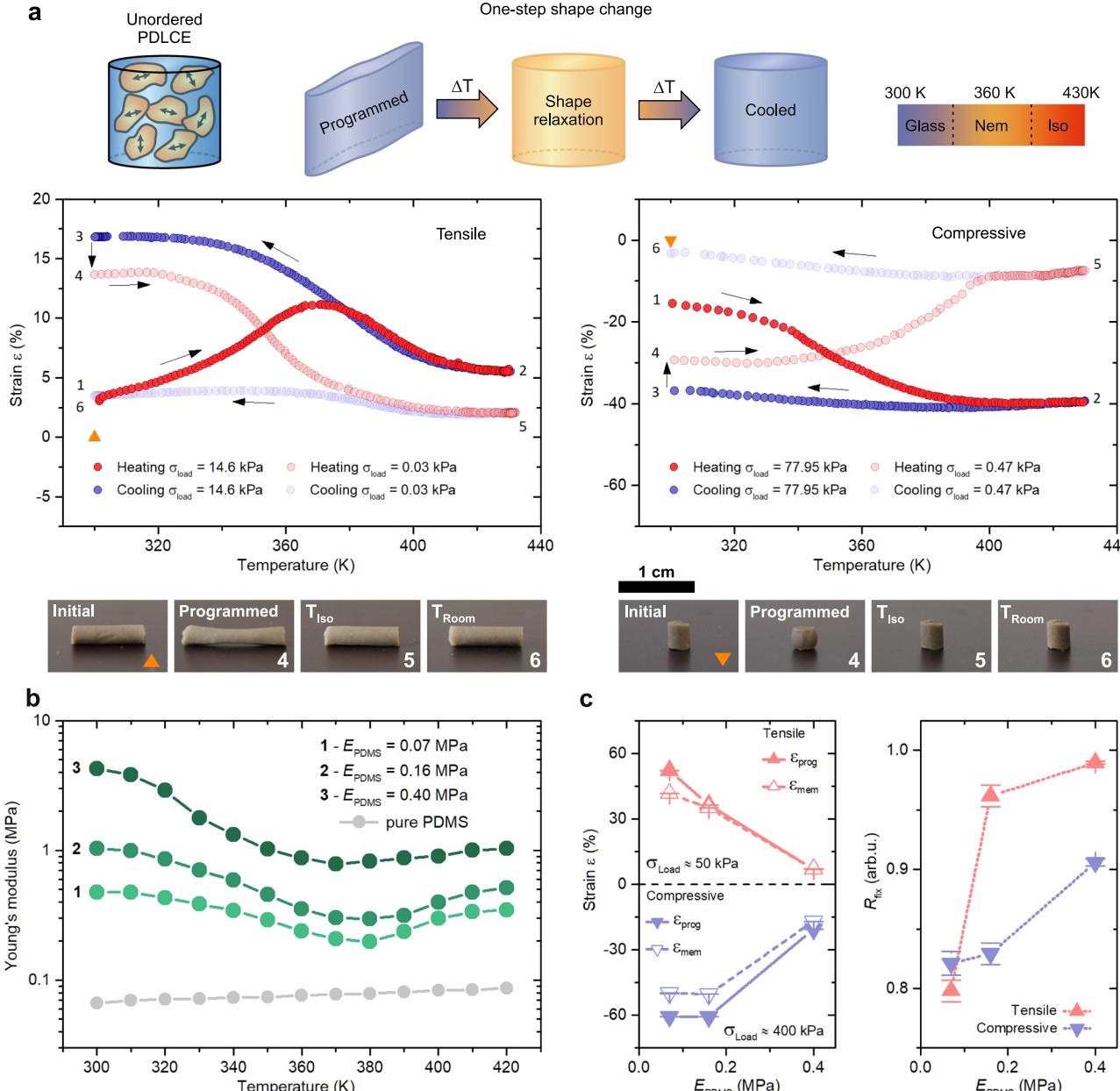

**Fig. 3 | Shape-memory programming and mechanical characteristics of PDLCE composites. a** Unordered PDLCEs exhibit a one-step shape change relaxation after heating above $T_{Glass-Nem}$. A complete strain relaxation occurs in the nematic phase ($T \approx 370$ K) in the case of tensile deformation (left graph - heating curve between points 4 and 5), while programmed compressive strain fully disappears only after $T_{Nem-Iso}$ (right graph). A slight TM contraction of $\lambda_{PDLCE} \approx 3\%$ is observed in both cases, typical for PDLCEs with no particle orientational order[33,39]. Bottom images show the measured sample at each stage of the thermal cycle presented in the graphs. **b** When compared to the bulk MC-LCE material, the $E_{PDLCE}(T)$ values appear much lower and the anomalies associated with individual LC transitions are less pronounced. Increasing the hardness of PDMS (light to dark green data) also increases the general $E_{PDLCE}(T)$ values of the composite. Pure PDMS sample (gray data) exhibits a typical linear thermal expansion. **c** Differences between the programmable ($\varepsilon_{prog}$, solid triangles) and memorized ($\varepsilon_{mem}$, empty triangles) strain can be observed after the programming thermal cycle (left graph). These are described by the fixation ratio $R_{fix}$ (right graph). A softer matrix seems to decrease the amount of memorized strain due to higher elasticity. Error bars were determined from three consecutive measurements.

recovers close to its initial length via elastic relaxations, and then contracts due to the thermally-induced Nem-Iso phase transition. Conversely, only a broad, one-step relaxation is again observed in the case of compressive stress programming (Fig. 5d, right graph).

The additional TM actuation offers new opportunities to introduce more intricate shape-changing abilities into a PDLCE specimen. A variety of morphing configurations can be imprinted into a single geometry by combining layers of differently aligned TM active and TM inert composite materials[33]. In combination with a lower temperature triggered SM relaxation, the PDLCEs exhibit a one-way shape change

between three distinct geometries, as well as actuate between the initial and magnetically imprinted shape when the SM strains are reset.

We demonstrate some of the specimen's morphing capabilities in (Fig. 6). A conventional 2D specimen can be designed to change shape in an in-plane fashion (Fig. 6, left column), morphing in sequence through three different 2D shapes. In this basic configuration, the particles are magnetically-aligned along the surface of the molded disc, which is then reshaped into a square. Upon heating, the programmed square shape relaxes to a circle and then contracts to an ellipse. Here, the bulk amount of the deformation occurs in

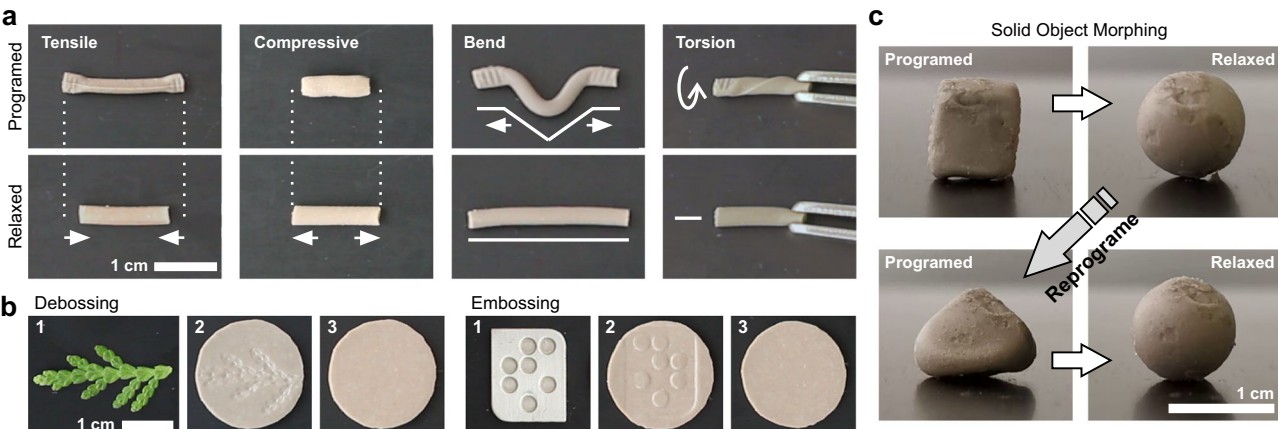

**Fig. 4 | Shape-programmability with various deformation modes. a** A cylindrical PDLCE shape is programmed by applying engineering stress during a thermal cycle, which runs from 450 K to 300 K (upper panels). The imprinted deformations reset to the initial shape after another thermal cycle (bottom panels). **b** The composite's surface can be patterned with a die or an object to produce stable and resettable embossing or debossing reliefs. **c** PDLCEs can be cast into solid 3D objects, which can be programmed multiple times into various shapes. This is demonstrated by reprogramming a PDLCE sphere into a cube or a tetrahedral. During thermal reset, the composite exhibits morphing back to its original spherical shape. All images were taken at room temperature. Please see Supplementary Movie 1–4 for detailed programming and shape-relaxation process.

two-dimensions, which makes such geometries suitable for applications where lateral actuation is needed, such as in microfluidics[50]. Note that the magnetically imprinted shape change is reversible, so upon the programmed strain recovery, the material can freely actuate between the molded and TM deformed geometry. Such deformation sequence could be utilized as a release and regulatory mechanism, for instance, working as a valve for initiating and then controlling the flow through a channel or a tube.

The same disc-shaped geometry can also be designed and programmed to perform out-of-plane morphing (Fig. 6, middle column). A bilayer disc sample is chosen here, made from a TM inert bottom and an ordered top layer with imprinted particle alignment along the sample's surface. Such a composite exhibits an out-of-plane bending actuation when heated above $T_{\text{Nem-Iso}}$. The sample shown here is programmed into a cone-like geometry, which, with increasing temperature, first relaxes downward into a flat circle and then lifts the two sides vertically to the particle's alignment via TM contraction of the upper layer. The transformation occurs in a '3D-to-2D-to-3D' fashion, but the morphing sequence can be rearranged by changing the molded and programmed geometries and the direction of the magnetic alignment.

The final example (Fig. 6, right column) is a PDLCE composite molded into a solid sphere and magnetically ordered so that the particle alignment is oriented along the sphere's vertical axis (pointing upward in the images). The sample is then reshaped into a cube. As a result, the sample morphs from a cube into a sphere; it then further contracts into an ellipsoid. The shape-change sequence occurs strictly between bulk, 3D solid objects and does not rely on bending deformations to attain a voluminous shape, like in the case of out-of-plane morphing.

## Discussion

We report on the utilization of the intrinsic, temperature-induced hardening of MC-LCEs, to functionalize a conventional silicone rubber with thermally-driven shape memory capabilities. A PDLCE composite material is prepared from a cured dispersion of MC-LCE microparticles dispersed in a PDMS matrix. Such composites are shown to exhibit effective shape-memory properties driven by the temperature-governed stiffness of the inclusions, which in turn prevents or promotes relaxations of the deformed matrix. The pre-polymerized PDLCE resin cures without volume change, which makes it practical to mold into arbitrary shapes and sizes, most significantly into bulk, solid 3D objects. Such fully filled specimens offer much greater

freedom in shape-programming in comparison with the conventionally molded 2D SM materials and their 3D-reshaped arrangements. For instance, compressive stress can be applied in all directions to reprogram a solid 3D object into unique 3D morphing geometries, which are difficult to attain by folding or reshaping a 2D system. Most importantly, the mechanical properties of the initial PDLCE material, e.g. Young's modulus, are retained by the shape-manipulated 3D objects, due to their fully filled volume with the shape-memory material. This is not the case for bending or folding actuations, where the mechanical response becomes a complex function of the 2D to 3D morphing topology. The intrinsic mechanical strength of fully filled objects thus eliminates the need for complex forming or layering of soft materials to achieve the desired structural resiliency.

Moreover, by magnetically aligning MC-LCE microparticles prior to curing the surrounding matrix, an additional TM reversible response can be imprinted into the composite. A magnetically-ordered PDLCE can therefore exhibit two shape changes with increasing temperature; the initial relaxation of the programmed strain and the second reversible actuation via collective LCE microparticle contraction when above $T_{\text{Nem-Iso}}$. Different combinations of programmed SM and imprinted TM actuation lead to a wide variety of possible shape-morphing configurations. Together with the overall facile handling, machining and low-demanding molding production process, PDLCEs serve as a practical shape-morphing material for straightforward implementation into future applications.

## Methods

### Monomer synthesis

Divinylic mesogenic monomers (MC-LCs) were synthesized using DCC-DMAP esterification between 2,3,5-trimethyl-hydroquinone ($M_{\text{qui}} = 152.19$ g mol$^{-1}$) and two units of 4-(3-butenyloxy) benzoic acid ($M_{\text{acid}} = 192,2$ g mol$^{-1}$), as per the procedure described in[51]. In detail, 4-(3-butenyloxy) benzoic acid (45.4 mmol, 10.49 g), 2,3,5-trimethyl-hydroquinone (22.7 mmol, 4.16 g), DCC (49.94 mmol, 12.39 g) and DMAP (4.54 mmol, 0.67 g) were dissolved in 130 ml of dichloromethane at room temperature and stirred for 72 h. The products were purified with column chromatography (CH$_2$Cl$_2$, SiO$_2$), with the first fraction separated and dried to obtain a white solid. The average yield was 79%, calculated from the percentage ratio between the mass of reactants and the mass of obtained in the first fraction (final product): %yield = (($m_{\text{qui}} + m_{\text{acid}}$)/$m_{\text{obtained}}$) × 100%. The powder was additionally filtered with ethanol to remove any impurities. All chemicals

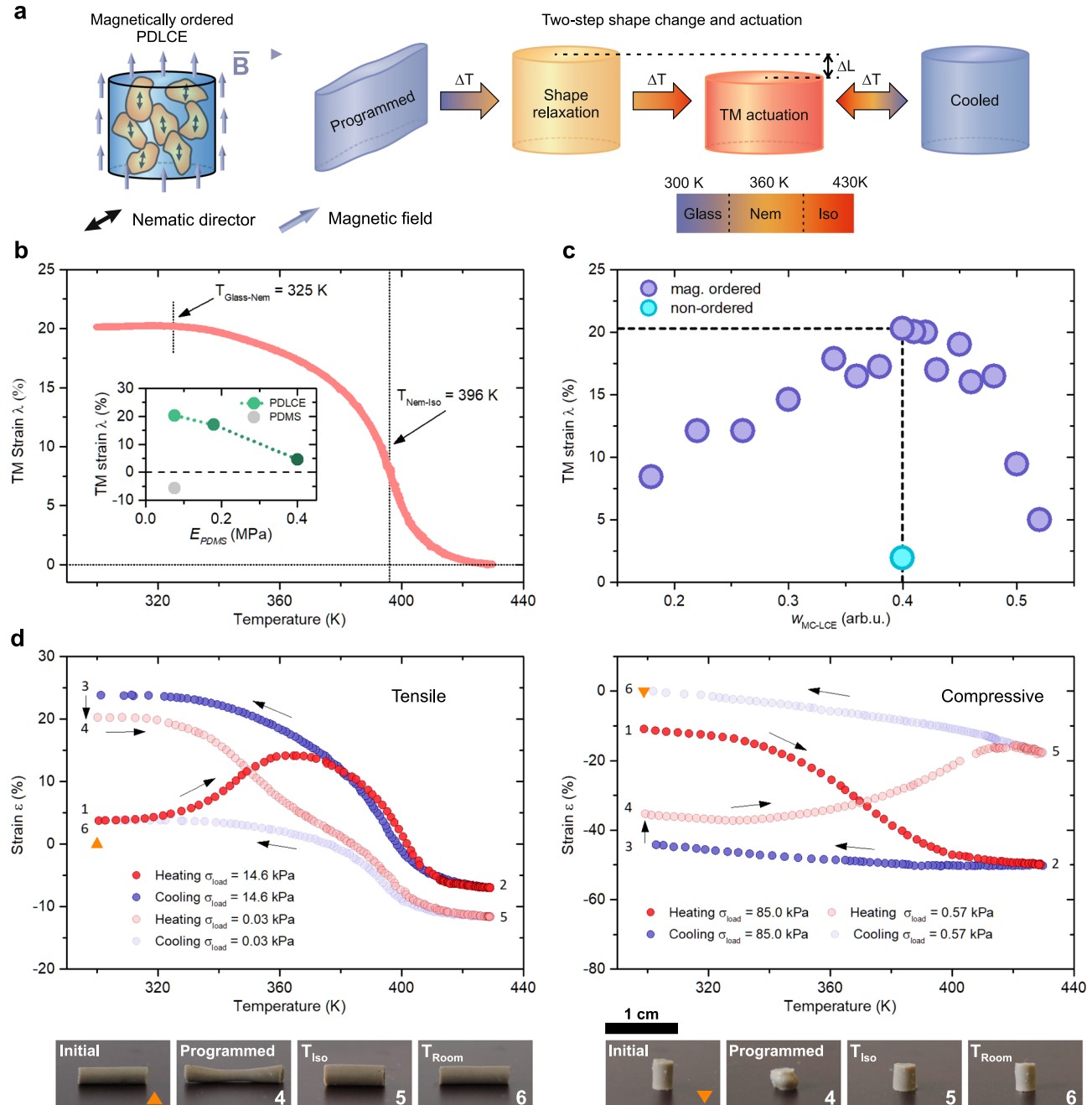

**Fig. 5 | Shape memory and thermomechanical actuation of PDLCE composites.**
**a** Magnetically oriented PDLCEs exhibit an additional TM contraction when above
$T_{Nem-Iso}$. Shape memory relaxation and TM actuation are well separated, with
increased temperature leading to two observable shape changes. The pre-
determined TM actuation is reversible with temperature; therefore, the composite
can morph between two geometries even after the shape memory is erased. **b** The
TM measurement almost completely mimics that of a bulk MC-LCE, with the
observable phase transitions slightly shifted to higher temperatures. For optimum
actuation (inset graph), $E_{PDMS}$ should not exceed $E_{LCE}$, so that the TM deformation

of LCE particles is not suppressed by the matrix' elasticity[33]. Gray data represent
measurements of the PDMS matrix material that was used for the preparation of the
standard composite samples. **c** The TM output is substantially reduced by the TM
inert PDMS matrix and can be controlled by the amount of MC-LCE particle
inclusions (purple circles). The maximum $\lambda_{PDLCE} = 20.3\%$ can be found at $w_{MC-}$
$_{LCE} = 0.40$ particle weight percentage. Teal circle indicates the response of a TM
inert composite with $\lambda_{PDLCE} \approx 3\%$. **d** Programming with tensile and compressive
stress (left and right graphs, respectively) is similar as in the TM inert case, except
for the large TM induced strain present during every thermal-cycle.

were purchased from Sigma Aldrich and used as received. The
obtained mesogenic monomers were characterized with ¹H NMR
(Supplementary Fig. 9) and ¹³C NMR, recorded with Bruker Avance
DRX 400 spectrometer.

¹H NMR (CDCl₃, 400 MHz): δ = 2.16 (s; 6H); 2.19 (s; 4H); 2.62 (m;
4H); 4.14 (m; 4H); 5.21 (m; 4H); 5.94 (m; 2H); 6.94 (s; 1H); 7.03 (m; 4H);
8.21 (m; 4H) ppm.

¹³C NMR (CDCl₃, 400 MHz): δ = 12.8; 13.2; 16.3; 33.2; 67.3; 76.9;
114.2; 117.5; 121.5; 127.8; 128.5; 130.8; 132.1; 134.0; 144.9; 146.5; 163.2;
164.0; 164.8 ppm.

**Elastomer synthesis**
Main-chain liquid crystal elastomers were prepared with two different
procedures: as a monodomain strip-shaped specimen synthesized via

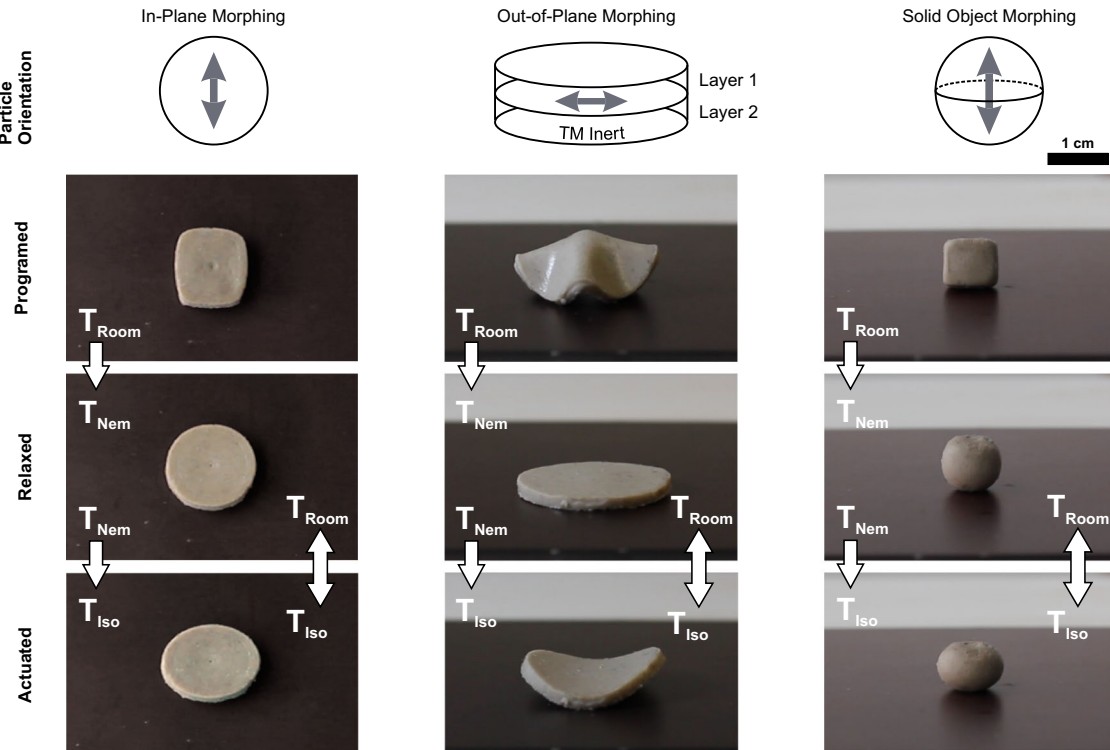

**Fig. 6 | Morphing capabilities of PDLCEs.** Three different morphing scenarios showcase the possible spatial shape-changing modes of a PDLCE. These modes are 'In-plane morphing' (left column), 'Out-of-plane morphing' (middle column) and 'Solid object morphing' (right column). The specimens exhibit shape changes from the programmed to the initial (relaxed) geometry once heated above $T_{\text{Glass-Nem}}$, and to the actuated state when above $T_{\text{Nem-Iso}}$ (from top to bottom). Once the programmed strains are reset, the specimens can still morph between the actuated and relaxed shape by heating and cooling them past $T_{\text{Iso}}$. Please see Supplementary Movie 5–7 for the complete morphing sequence.

the two-stage crosslinking procedure[38], or as a bulk specimen, where the second crosslinking stage is omitted and the LCE solution is polymerized in the presence of an external magnetic field, producing a partially ordered MC-LCE material[33,39].

Both methods include dissolving 1,1,3,3-Tetramethyldisiloxane (1.2 mol) and 2,4,6,8,10-Pentamethylcyclopentasiloxane crosslinker (0.08 mol) against 1 mol of mesogenic monomers in anhydrous toluene (1.3 ml). The MC-LCE solution is heated to 343 K to completely dissolve all components and (1.5-cyclooctadiene)platinum dichloride dissolved in dichloromethane (71 mmol l$^{-1}$) platinum catalyst solution (30 μl) is added.

For producing strip-shaped monodomain MC-LCEs, the prepared concoction is filtered into a home-built cylindrical form and centrifuged at 970 x g at 343 K for 48 h. The obtained partially crosslinked MC-LCE film is removed, cut into smaller similarly-sized strips and strained using several 3 mg weights until the specimen becomes completely transparent, indicating a predominately monodomain structure. Strained samples are further crosslinked at 343 K for another 24 h. Typical dimensions of MC-LCE strips were 20 mm in length, 3.20 mm in width and 0.25 mm in thickness.

Partially oriented bulk MC-LCE samples were prepared in PTFE flasks filled with MC-LCE solution and crosslinked overnight at 343 K inside a wide bore superconductive magnet (Bruker Avance III 500 MHz) with a homogeneous magnetic field of $B = 11.74$ T. The magnetic field facilitates greater ordering of the mesogenic components, promoting the growth of bigger nematic domains. This ensures that the MC-LCE microparticles obtained by freeze-fracturing of the partially ordered LCE material are predominately monodomain[33,39]. As such, the mesophase behavior and TM contraction of the parent MC-LCE material are preserved[40]. A cube with $2.3 \pm 0.2$ mm sides was cut from the homogeneous section of the

synthesis for thermal-cycling under applied compressive stress experiments.

### PDLCE composite preparation

Bulk-synthesized, partially ordered MC-LCE material was solely used for the preparation of PDLCEs. The material was cut into ~1 mm$^3$ pieces and mixed with PDMS (polydimethylsiloxane−Sylgard® 184 Silicon Elastomer Kit) in a 1:1 weight ratio. The mixture was freeze-fractured using a cryogenic mill CryoMill by Retsch, GmbH, with pre-cooling at 5 Hz for 2 min, followed by three milling cycles at 30 Hz for 12 min, conducted in 3-min intervals that were separated by a 30 s intercooling cycle at 5 Hz. The resulting LCE particle size distribution is between 0.4 μm and 25 μm[40]. The obtained PDLCE melt is then heated to 363 K and mixed to acquire a homogeneous dispersion.

For molding of the composites, the suspension is first reduced to the desired concentration and the appropriate amount of PDMS curing agent is added. The 1:40 curing agent to PDMS weight ratio has been considered as optimal for achieving the largest TM output. The mixture is then evacuated, introduced into PTFE molds and cured overnight at 343 K with or without the presence of an orientational magnetic field, in order to produce TM active or TM inert composites, respectively.

### Mechanical measurements

All mechanical measurements have been performed on a home-made extensometer. The experimental setup consists of a heating chamber, with the observed sample positioned in the center. The sample is attached between two metal holders, with the bottom holder fixed to a strain gauge and the upper holder to a translation stage. The load is measured in dependence to the position of the translation stage.

The thin strip LCE samples were clamped between the two metal holders, while the cylindrically-shaped composites were attached to the holders with epoxy glue to eliminate unwanted sample deformation from clamping. Epoxy glue has a significantly larger Young's modulus (several GPa) than our samples ($E_{max} = 3$ MPa), so no significant impact of the epoxy layer on mechanical measurements is expected. All PDLCE samples used in experiments performed with the extensometer were cylindrically-shaped, with typical dimensions of 20 mm in length and 2.1 mm in diameter. The dimensions for the strip-shaped and the partially magnetically oriented cube-shaped MC-LCE samples were the same as the dimensions obtained from the synthesis. The nematic director in the investigated TM active samples was always parallel to the applied mechanical stress.

Thermomechanical investigations were conducted by applying a constant load of $\sigma_0 = 0.6$ kPa on the sample and cooling the specimen from 430 K to 300 K with the rate of 0.5 K min$^{-1}$. Only the cooling curve was recorded to avoid any leftover strain originating from the sample handling, which can influence the measurements during the heating run.

Temperature-dependent, stress-strain measurements were performed within the temperature range of 300 K to 430 K, with a temperature step of 10 K. Stress-strain curves were recorded by applying stress from $\sigma_0 = 0.6$ kPa to $\sigma_{max} = 37$ kPa for LCEs and from $\sigma_0 = 0.6$ kPa to $\sigma_{max} = 8.5$ kPa for PDLCEs. Measurements were fitted with $\sigma = \sigma_0 + E \times \varepsilon$ to yield the Young's modulus.

The shape-memory capabilities of PDLCE composites were examined by heating and cooling the sample (thermal cycling), with and without an applied load, in the form of either tensile or compressive stress. Samples were initially loaded with a constant load until no more change in strain was observed, after which they were thermally cycled while measuring the sample's length. The load was then removed and the relaxation of imprinted strain was measured in order to determine the degree of strain fixation. Another thermal cycle under zero-limit applied load was performed to reset the memorized deformation. Shape-memory tests were run between 300 K to 430 K. The same cylindrical samples as for other experiments were used in the tensile stress mode, with applied load of $\sigma_{load} \approx 15$ kPa for programming and $\sigma_{zero} < 0.2$ kPa for the zero-load limit. In the compression stress mode, the specimens' dimensions were 2.2 mm in height and 2.1 mm in diameter and a higher programming and zero-limit compressive stress of $\sigma_{load} \approx 80$ kPa and $\sigma_{zero} < 0.6$ kPa was applied. Additional thermal-cycling measurements with higher applied stress were performed at $\sigma_{load} \approx 50$ kPa and $\sigma_{zero} < 0.2$ kPa for tensile and $\sigma_{load} \approx 400$ kPa and $\sigma_{zero} < 10$ kPa for compressive stress mode.

## Calorimetry

The phase transition behavior of both the divinylic mesogenic monomer and the MC-LCE sample has been measured by means of ac calorimetry. The experimental apparatus is fully automated, home-made at the Jožef Stefan Institute (Slovenia). It operates with slow scanning rates and provides a very precise temperature dependence of the heat capacity[52,53]. A sample quantity of 30 mg has been placed in high-purity silver cells, with a heater and a thermistor attached to two sides. Two additional shields are used, the outer one being immersed in a vessel filled with thermal oil in order to achieve temperature stability of ~ 0.1 mK. After being mounted to the calorimeter, the sample was heated to the isotropic phase and cooled down to 370 K with a scanning rate of 0.3 K h$^{-1}$ and then further to room temperature with a scanning rate of 0.5 K h$^{-1}$.

## Data availability

Raw data generated in this study is available at https://doi.org/10.6084/m9.figshare.21953051. Materials and additional data are available from the authors.

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

## Acknowledgements

This work was supported by the Slovenian Research agency (ARRS), applied project L1-2607 (A.R.) and research program P1-0125.

## Author contributions

A.R. defined the initial research directions. V.D., S.U. and A.R. synthe-sized main-chain mesogens components and liquid crystal elastomers. M.B and A.R. prepared polymer-dispersed liquid crystal elastomer composites. M.B., A.R., N.D., N.N., M.L. and G.C. performed the experi-ments. A.R. and B.Z. supervised the study and analyzed experimental data. M.B. prepared graphics and movies. A.R. wrote the paper, with contributions of all authors.

## Competing interests

The authors declare no competing interests.
