## [Peer Review File · Nature Communications]

Polymer-dispersed liquid crystal elastomers as moldable shape-programmable materialReviewers' Comments:

Reviewer #1:

Remarks to the Author:

Bobnar et al. describe the shape morphing and irreversible shape memory of liquid crystal elastomer - elastomer composites. This work builds on a prior report, which showed that LCE microparticles within an elastomeric matrix enables reversible shape change. In this paper, the glass transition of the LCE microparticles is used to induce a one-way shape memory effect.

Building composites where LCEs are the dispersed phase is original (beyond one prior publication from the authors). The conclusions of the work are well supported by the data.

The new component of the manuscript focuses on the irreversible behavior of the composites. This behavior is derived from the change in the modulus of the microparticles. Such behavior is fairly widely reported in the literature. <https://doi.org/10.1016/j.jmps.2011.09.011>

In summary, the paper reports an interesting system of composites that are somewhat unique. However, the new feature in this publication is based on very well-established techniques to create irreversible shape memory in composites. This paper will be of mostly of interest to the specialist. As a result, I think that this work would be best suited for a field-specific journal.

Reviewer #2:

Remarks to the Author:

The authors present a composite liquid-crystal based material capable of interesting shape morphing properties. These materials are very different from those typically studied in the LCE field since they contain particles of a main-chain LCE embedded in a flexible silicone elastomer. The authors claim that "Until now, full-bodied SM systems of this kind were attained only with additive manufacturing processes" but I do not agree. There are several examples of LCEs capable of similar shape-changes. While I do agree that the composites are easier to mold and shape compared with most LCEs, prior works have demonstrated that LCEs can be molded, embossed, and programmed into complex shapes.

Overall the work is clear, the authors do an excellent job describing their material and the unique shape-morphing properties. However, I think they overlook work with LCEs that can achieve similar shape changes. For example, DOI 10.1039/C8SM02174K presents many examples of 3D to 3D shape changes using LCEs. Another example is 10.1073/pnas.191795211. There are other studies demonstrating complex shape changes in LCEs, and I don't see how the PDMS composites reported enable actuation that is not possible with these LCE systems.

A clear novelty of the materials is in being able to access a glass transition and nematic-to-isotropic transition temperature. The authors demonstrate the advantages of this in their work. They also very clearly describe the mechanical properties and shape programming.

Overall this is an excellent study, and I recommend that the authors properly put their work in context of other shape-morphing LCE materials.

Reviewer #3:

Remarks to the Author:

The manuscript "Polymer-dispersed liquid crystal elastomer as moldable shape-programming material" presents a versatile strategy to introduce shape morphing between 3D shapes, which is vital important

and hard-to-achieve for liquid crystal elastomers. I actually was thinking how to realize this kind of shape morphing during the last several years. And I was very excited when reading this manuscript for the first time. In addition, the manuscript is well written and I really enjoy reading this manuscript. Due to the importance of the demonstrated capability, I think this manuscript should be published in Nature Communications. And I would definitely recommend publication of this manuscript!

I suggest authors to consider some minor revisions:

1. I suggest to introduce current development of LCE shape morphing in the introduction to show the importance of this work. For example, 2D flat films to arbitrary surface geometries (PNAS 2018, 115, 7206-7211), bending deformation of 3D printed structures (ACS Appl. Mater. Interfaces 2019, 11, 28236-28245), programmable shape morphing of 3D assembled frames (Nature Comm. 2021, 12, 5936), and others.

2. I find that authors did not mark (a), (b), (c), and (d) correctly in Figure 1. Please change this figure and read through the manuscript to correct the errors.

Reviewed by Yubing Guo, Beijing Institute of Technology, China

Response to referees' comments

Reviewer #1:

We thank the Referee for positive evaluation of our work and carefully address the concern regarding the suitability of the manuscript for a field-specific journal.

Comment 1 - The new component of the manuscript focuses on the irreversible behavior of the composites. This behavior is derived from the change in the modulus of the microparticles. Such behavior is fairly widely reported in the literature. <https://doi.org/10.1016/j.jmps.2011.09.011>
In summary, the paper reports an interesting system of composites that are somewhat unique. However, the new feature in this publication is based on very well-established techniques to create irreversible shape memory in composites. This paper will be of mostly of interest to the specialist. As a result, I think that this work would be best suited for a field-specific journal.

Author's response –

We do not claim that the mechanism behind the shape-memory capability of our composites is specifically new or unique. We rather report on using shape-memory inclusions as means of functionalizing silicone elastomer with shape-morphing capabilities and exploiting the particulate nature of the inclusions that enables us to produce specimens of different shapes and sizes, especially the ability to mold the dispersion into bulk, full-bodied shape-morphing systems, which we also fully demonstrate in the paper.

We acknowledge that a number of similar composite systems have been reported in literature, with focus mostly on synthesis methods. In the vast majority of cases, the final result is limited to 2D specimens, with the exception of additive manufacturing. As an example, in the work provided as an example, i.e. the preparation of silicone shape-memory composites from an electro-spun network (<https://pubs.acs.org/doi/10.1021/ma9015888>), the reported method can indeed be, in principle, used to produce larger voluminous samples. However, the geometries of the composite material are limited to shapes cut from the removed electro-spun flat mesh, while the yield of the material synthesis is not ideally suited for upscaling. This limits the practical implementation of such composites.

For instance, the wide-spread use of silicone elastomers is largely attributed to their ease of manufacturing and molding. This is hardly ever true for conventional shape-memory materials or similar composites whose synthesis relies on complex chemistry or other specific processes. On the contrary, our material can be straightforwardly formed into bulk, 3D solid objects, which is its main advantage. We specifically describe this in the introduction and throughout other parts of the manuscript. Therefore, we do not share the same opinion with the Reviewer that this work would mostly be of interest to the specialists, on the basis that such a shape-memory mechanism has already been proposed. The properties and the demonstrated capabilities of the investigated composite material go beyond this claim, offering a new and versatile approach towards 3D morphing, interesting for the broad audience of Nature Communications.

We do acknowledge that the context in some sections of the paper might have not been clear. Therefore, we have changed a few parts of the manuscript's text and added additional references to avoid any possible misinterpretations by the readers.

These changes are described below:

Correction 1:

The abstract has been partially rewritten to give greater emphasis on the moldability feature of the composite material. The new abstract thus starts with:

'Current development of soft shape-memory materials often results in novel materials that are typically limited to the synthesis of thin-walled specimens or have to rely on complex, usually low-yield manufacturing techniques to fabricate macro-sized, solid three-dimensional objects. Such geometrical limitations and slow production rates can significantly hinder their practical implementation. In this work, we demonstrate a shape-memory composite material that can be effortlessly moulded into arbitrary shapes or sizes. The composite material is made from main-chain liquid crystal elastomer (MC-LCE) microparticles dispersed in a silicone polymer matrix. Shape-programmability is achieved via low-temperature ...'

Correction 2:

Text on Page 3, Paragraph 2, Line 85 was changed to:

'In the present study, we utilize the intrinsic glassiness of nematic MC-LCE microparticles, rather than their TM actuation capabilities, to functionalize the composite material with reprogrammable shape-memory that resemble those of SM polymers and related composites.'

with added additional citations focusing on relevant shape-memory composites ([Xia, Y. et. al. **2021**], [Luo, X. et. al. **2009**] and [Ge, Q. et. al. **2012**] numbered [9], [34], [35], respectively).

Reviewer #2:

We are happy for the Referee's positive evaluation of our work and thankful for the comments how to further improve the manuscript. Our response can be found below.

Comment 1 - The authors claim that "Until now, full-bodied SM systems of this kind were attained only with additive manufacturing processes" but I do not agree. There are several examples of LCEs capable of similar shape-changes. While I do agree that the composites are easier to mold and shape compared with most LCEs, prior works have demonstrated that LCEs can be molded, embossed, and programmed into complex shapes.

Author's response – Although such molding methods are regularly applied for the production of LCEs and SM polymers, the molding is usually performed on highly swollen systems, which upon drying do not retain the shape of the mold. Similarly, the final geometry of the material synthesized with photocuring is greatly limited by the short penetration length of UV light. Most shape-memory materials are therefore restricted to 2D thin specimens unless produced in microscopic geometries or with additive manufacturing, where the sequential deposition of small quantities of material also resolves these issues. Nevertheless, we agree with the Reviewer that the aforementioned claim does not precisely reflect this and has been therefore removed. In order to clearly state the above written limitations, we have also changed parts of the introduction section, as explained below.

Correction 1:

The sentence on Page 3, Paragraph 2, Line 95, 'Until now, full-bodied SM systems of this kind were attained only with additive manufacturing processes.' has been removed.

Correction 2:

Part of the previously written Paragraph 3 on Page 2, Line 62, that read 'These usually require vast amounts of solvent that needs to be removed once the reaction is completed, or rely on light-induced polymerization and other similar processes that limit any larger-scale fabrication of SM materials to thin-film geometries.' has been changed to:

'These typically require vast amounts of solvent that needs to be removed once the reaction is completed. Molding is therefore performed on swelled systems, which upon drying do not retain the initial geometry of the mold. Photo-curing is also commonly used for the production of SM material, but the final geometries are limited by the short penetration lengths of UV-light. Together with other similar methods, any larger-scale fabrication and molding of SM materials remains generally restricted to thin-film geometries.'

Comment 2 - However, I think they overlook work with LCEs that can achieve similar shape changes. For example, DOI 10.1039/C8SM02174K presents many examples of 3D to 3D shape changes using LCEs. Another example is 10.1073/pnas.191795211. There are other studies demonstrating complex shape changes in LCEs, and I don't see how the PDMS composites reported enable actuation that is not possible with these LCE systems.

Author's response – Our response to this comment also includes remarks to the previous comment where the reviewer states that many shape-memory materials can also be embossed and programmed into complex shapes;

While some interesting examples of a 3D to 3D shape changes do exist, these are usually performed with 2D sheets or a 3D arrangement made of 2D sheet segments, resulting in bending, origami-like actuations or more complex morphing into hollow objects. In our case we deal with a real 3D to 3D reshaping and, more specifically, with a compressive stress that can be applied in all directions, offering much greater freedom in shape-programming. This is much more difficult to achieve by folding or reshaping 2D sheets. We demonstrate this by comparing the morphing mechanism of 2D flat-shaped samples, exhibiting the more 'conventional' 3D morphing, with the shape-transitions of solid shape specimens. Similarly, the most commonly reported embossing/debossing is not performed by compressing the material with a die to alter its thickness (unless on a microscopic scale), but rather by pressing/bending a 2D sheet into or around the die to produce a 3D relief.

We again emphasize that the originality of this work is the solidness of the samples. Prepared 3D geometries are homogenously-filled objects which retain their mechanical properties (e.g. Young's modulus) regardless of the programmed shape. This is not the case for the origami style approach, where the mechanical response becomes a complex function of 2D to 3D folding topology.

We nevertheless understand that the general term '3D to 3D' describes 3D shape-changes and is not related to the morphing mechanisms based on the initial nature of the specimen's geometry, with which we established our terminology in the manuscript. In order to clarify our arguments and make our claims more precise, we have rewritten parts of the text.

The detailed changes are given below:

Correction 1:

We discuss some of the advantages of bulk 3D objects over the conventional 2D systems in the newly written Discussion section of the manuscript. This includes the next argument:

Page 16, Paragraph 2, Line 432:

'The pre-polymerized PDLCE resin cures without volume change, which makes it practical to mold into arbitrary shapes and sizes, most significantly into bulk, solid 3D objects. Such fully filled specimens offer much greater freedom in shape-programming in comparison with the conventionally molded 2D SM materials and their 3D-reshaped arrangements. For instance, compressive stress can be applied in all directions to reprogram a solid 3D object into unique 3D morphing geometries, which are difficult to attain by folding or reshaping a 2D system. Most importantly, the mechanical properties of the initial PDLCE material, e.g. Young's modulus, are retained by the shape-manipulated 3D objects, due to their fully filled volume with the shape-memory material. This is not the case for bending or folding

actuations, where the mechanical response becomes a complex function of the 2D to 3D morphing topology. The intrinsic mechanical strength of fully filled objects thus eliminates the need for complex forming or layering of soft materials to achieve the desired structural resiliency.'

Correction 2:

Several changes were made to the naming of the morphing mechanisms on Page 14 and 15, in Figure 6 and its associated caption. The term '2D morphing' is changed to 'In-plane morphing', 'Quasi 3D morphing' to 'Out-of-plane morphing' and 'Real 3D morphing' to 'Solid object morphing'.

On account of these changes, several sentences were rewritten or omitted in the same paragraphs. These are:

Sentence on Page 14, Paragraph 3, Line 382, 'We demonstrate some of the morphing capabilities in (Figure 6), based on spatial TM transformation.' now reads:

'We demonstrate some of the morphing capabilities in (Figure 6)'

Sentence on Page 15, Paragraph 2, Line 395, 'The same disc-shaped geometry can also be designed and programmed to perform quasi 3D morphing, i.e. an out-of-plane shape change of a 2D specimen into 3D space (Figure 6, middle column).' now reads:

'The same disc-shaped geometry can also be designed and programmed to perform out-of-plane morphing (Figure 6, middle column).'

Sentence on Page 15, Paragraph 3, Line 409, 'We consider this as real 3D morphing' was removed.

Sentence on Page 15, Paragraph 3, Line 409, 'The shape-change sequence occurs strictly between 3D geometries and does not rely on 2D-to-3D deformations to attain a voluminous shape.' now reads:

'The shape-change sequence occurs strictly between bulk, 3D solid objects and does not rely on bending deformations to attain a voluminous shape, like in the case of out-of-plane morphing.'

Correction 3:

Other minor corrections were done on parts of the manuscript pertaining to the production and morphing of PDLCEs:

The sentence on Page 3, Paragraph 2, Line 99 that reads 'Combined with the SM programmability, the material can be formed into various shape-morphing configurations that truly exhibit 3D morphing, in contrast to the conventional 2D-to-3D bending-like deformations realized with the currently available SM materials.' has been changed to:

'Combined with the SM programmability, the material can be formed into various morphing configurations exhibiting a two-step, temperature-driven shape transformation.'

The sentence on Page 9, Paragraph 3, Line 260 that reads 'Because PDLCEs can be molded into full-bodied 3D objects, a compressive programmable stress can be used to imprint substantial deformations into the specimen (up to $\epsilon_{SM} = 49\%$, Supplementary Figure 8). Until now, out-of-plane programming of SM materials was typically limited to surface embossing of thin specimens, which resulted in micro-scaled responses.' has been changed to:

'Due to the typical 2D nature of SM materials, compressive stress programming of SM material is typically restricted to surface embossing of thin samples, which results in micro-scaled deformations of the material's thickness. Since PDLCEs can be molded into voluminous, solid 3D objects, a compressive programmable stress can be used to imprint substantial deformations into the specimen (up to $\epsilon_{SM} = 49\%$, Supplementary Figure 8).'

Paragraph 2 on Page 11, Line 297 was changed from '3D PDLCE samples with real morphing capabilities that go beyond the origami-like bending of conventional 2D SM systems can be produced. For instance, a solid spherical PDLCE can be programmed into a cube, which in turn morphs back into the original spherical shape once thermally reset (Figure 4c). The same PDLCE can then be reprogrammed into a new stable shape, in this case a tetrahedron, and reset when desired.' and merged with the next paragraph into:

'Bulk, three-dimensionally produced PDLCE samples can perform morphing between solid objects. For instance, a solid spherical PDLCE can be programmed into a cube, which in turn morphs back into the original spherical shape once thermally reset (Figure 4c). The same PDLCE can then be reprogrammed into a new stable shape, in this case a tetrahedron, and reset when desired. Here, shape-morphing occurs via relaxations of the material's deformed solid volume. Due to the general mechanical robustness of such solid systems ...'

Comment 3 - ... I recommend that the authors properly put their work in context of other shape-morphing LCE materials.

The manuscript has now been changed according to the previous two comments, which place it more appropriately and precisely in the context of relevant research.

Other changes:

The citation list was updated and now includes two additional citations from work mentioned in the reviewer's Comment 2. These are [Shahsavan, H. et. al., 2020] as citation [4] and [Barnes, M. et. al., 2019] as citation [8].

Reviewer #3:

We are happy for the positive evaluation of our work by the Reviewer, and have followed all the suggestions for minor revisions to improve the manuscript, as explained below.

Comment 1 - I suggest to introduce current development of LCE shape morphing in the introduction to show the importance of this work. For example, 2D flat films to arbitrary surface geometries (PNAS 2018, 115, 7206-7211), bending deformation of 3D printed structures (ACS Appl. Mater. Interfaces 2019, 11, 28236-28245), programmable shape morphing of 3D assembled frames (Nature Comm. 2021, 12, 5936), and others.

Author's response – We have updated our reference list and cited additional recent research to reflect the current status of LCE development.

In detail, these include:

[Shahsavan, H. et. al., 2020] as citation [4], [Aharoni, H. et. al., 2018] as citation [7], [Xia, Y. et. al. 2021] as citation [8], [Guo, Y. et. al., 2021] as citation [16] and [Tabrizi, M. et. al., 2019] as citation [32].

Comment 1 - I find that authors did not mark (a), (b), (c), and (d) correctly in Figure 1. Please change this figure and read through the manuscript to correct the errors.

Author's response – Figure 1 and the caption text has been corrected. Marks are now consistent with the text. We thank the Reviewer for bringing this detail to our attention.

Other significant changes to the manuscript:

Former 'Conclusions' section has been shortened and rewritten for the required Discussion section.

Terms like 'full-bodied', 'full-volume' and 'filled-volume' used to describe the non-hollow or solid objects has been substituted throughout the manuscript with the more appropriate term 'solid object' or 'solid 3D object'.